# Impact of Refrigerated Storage on Microbial Growth, Color Stability, and pH of Turkey Thigh Muscles

**DOI:** 10.3390/microorganisms12061114

**Published:** 2024-05-30

**Authors:** Agnieszka Orkusz, Giorgia Rampanti, Monika Michalczuk, Martyna Orkusz, Roberta Foligni

**Affiliations:** 1Department of Biotechnology and Food Analysis, Wroclaw University of Economics and Business, 53-345 Wroclaw, Poland; 2Dipartimento di Scienze Agrarie, Alimentari ed Ambientali, Università Politecnica delle Marche, Via Brecce Bianche, 60131 Ancona, Italy; g.rampanti@pm.univpm.it (G.R.); r.foligni@staff.univpm.it (R.F.); 3Department of Animal Breeding and Production, Warsaw University of Life Sciences, 02-786 Warsaw, Poland; monika_michalczuk@sggw.edu.pl; 4Faculty of Biotechnology and Food Science, Wroclaw University of Environmental and Life Sciences, 50-375 Wroclaw, Poland; 125360@student.upwr.edu.pl

**Keywords:** turkey, storage time, temperature, microbial growth, color, food safety

## Abstract

The quality of poultry meat offered to the consumer depends mainly on the level of hygiene during all stages of its production, storage time, and temperature. This study investigated the effect of refrigerated storage on the microbiological contamination, color, and pH of turkey thigh muscles stored at 1 °C over six days. Microbial growth, including total mesophilic aerobes, presumptive lactic acid bacteria, and Enterobacteriaceae, significantly increased, impacting the meat’s sensory attributes and safety. On the 6th day of meat storage, the content of total mesophilic aerobes, presumptive lactic acid bacteria, and Enterobacteriaceae was 1.82 × 10^7^ CFU/g, 1.00 × 10^4^ CFU/g, and 1.87 × 10^5^ CFU/g, respectively. The stability of color was assessed by quantifying the total heme pigments, comparing myoglobin, oxymyoglobin, and metmyoglobin concentrations, analyzing color parameters L*, a*, b*, and the sensory assessment of surface color, showing a decline in total heme pigments, three myoglobin forms, redness (a*) and lightness (L*). In contrast, yellowness (b*) increased. These changes were correlated with the growth of spoilage microorganisms that influenced the meat’s pigmentation and pH, with a notable rise in pH associated with microbial metabolization. Based on the conducted research, it was found that the maximum storage time of turkey thigh muscles at a temperature of 1 °C is 4 days. On the 4th day of storage, the total mesophilic aerobe content was 3.5 × 10^5^ CFU/g. This study underscores the critical need for maintaining controlled refrigeration conditions to mitigate spoilage, ensuring food safety, and preserving turkey meat’s sensory and nutritional qualities. There is a need for further research to improve turkey meat storage techniques under specific temperature conditions by studying the impact of using varying packaging materials (with different barrier properties) or the application of natural preservatives. Additionally, future studies could focus on evaluating the effectiveness of cold chain management practices to ensure the quality and safety of turkey products during storage. By addressing these research gaps, practitioners and researchers can contribute to developing more efficient and sustainable turkey meat supply chains, which may help mitigate food wastage by safeguarding the quality and safety of the meat.

## 1. Introduction

Among the various species of poultry meat, turkey meat is considered the leanest. Its dietary value is determined by the high protein level. Turkey protein has a unique amino acid composition. It is easily digestible and contains more tryptophan—an amino acid for which turkeys are susceptible—than the meat of other birds. Turkey meat is delicate and tender and does not require high temperatures or long cooking times during preparation [1].

The quality of meat is influenced by numerous factors, with color being a significant determinant. Meat color is a key factor influencing consumer decisions, with discoloration leading to significant revenue loss [2,3,4]. The variation in meat color is influenced by a multitude of intrinsic (such as sex, age of the animal, muscle type, and pH of the meat) and extrinsic (including temperature, oxygen availability, and type of microorganisms) factors [5], including the concentration and chemical state of pigments, and the physical structure of the meat [6,7]. The primary pigment accountable for the distinctive color of fresh meat is myoglobin, which exists in three forms: bright red oxymyoglobin (MbO_2_), purple-red deoxymyoglobin (Mb), and brown metmyoglobin (MMb) [8,9,10,11]. The reduction of metmyoglobin activity is a critical aspect in maintaining the quality of meat, as it directly impacts color stability. Metmyoglobin, a form of myoglobin that is oxidized and typically present in meat, undergoes a transformation into its reduced form, oxymyoglobin, during this process. Various internal factors, such as meat pH can influence the redox potential and consequently affect the efficiency of metmyoglobin reduction. External factors, including storage conditions and temperature, play a role in metmyoglobin production, with oxygen exposure and prolonged storage periods all contributing to undesirable color changes [12].

Food safety is a global issue that requires an integrated global response. The idea of maintaining food safety “from field (farm) to fork” is a strategic direction of the European Union’s (EU) actions; it is also a response to the expectations of consumers who are increasingly looking for high-quality products and knowledge about production standards and its stages. This is the so-called food chain, which is the control of animal sources and proper nutrition after slaughter, as well as packaging and transport [13]. Currently, there are two European Commission regulations; the first one is from 2004 [14], concerning the hygiene of food of animal origin, and the second one is from 2005, regarding microbiological criteria for foodstuffs [15]. It is assumed that the elements obtained from the division of carcasses can be stored at −1 to 2 °C for 48 h, while whole carcasses should be at −2 to 4 °C for up to 6 days.

Meat with a high water activity (0.97) and a pH below 6.0 is a favorable environment for developing microorganisms [16,17].

Except for *Samonella* spp., there is no precise microbiological requirement to be observed during the storage (shelf life) of poultry meat. Therefore, the overall number of bacteria at 6–7 log CFU/g is an important limit value for chilled meat [18,19].

In cases where the number of microorganisms reaches 7–8 log CFU/g, sensory signs of meat deterioration appear, i.e., changes in odor and color. Above this level, there are texture changes and slime on the meat. Furthermore, it has been shown that the microbial status of meat starts to deteriorate when the total microbial counts reach 7–8 log CFU/g [20,21]. This is supported by the fact that spoilage perception is subjective, and chemical indicators, including biogenic amines, have been proposed to assess meat quality and spoilage [22]. Moreover, the existence of specific microorganisms, like Enterobacteriaceae, and lactic acid bacteria, is intricately connected to the deterioration of refrigerated poultry, indicating a possible correlation between microbial counts and sensory alterations [23].

The impact of cold temperature storage on meat quality remains a topic of ongoing scientific investigation aimed at enhancing the product’s shelf life. Poultry meat is commonly marketed at refrigerated temperatures (2–5 °C) [24]. That is why most shelf-life studies for meat (mostly chicken) have focused on refrigerated temperatures above 0 °C, mainly 4 °C [24,25,26,27,28] or frozen temperatures at −18 °C [25,29]. Turkey and chicken meat differ due to the genetic background of the animals, which determines the meat’s structure and chemical composition. Exogenous factors such as farming, nutrition, transport, and slaughter conditions also play a role in these differences. The reason for the differences in the biochemical properties of turkey and chicken meat is believed to be the different glycolytic potential, which affects the pH values, as well as the oxidative potential, which affects the color of the meat by changing the variants of oxymyoglobin, deoxymyoglobin and metmyoglobin [30]. Since there is a lack of studies in the available literature regarding the storage of meat at 1 °C, and it cannot be conclusively stated that changes in turkey meat will occur in the same way as in chicken meat, the objective of this work was to compare the changes in the microbiological contamination, color, and pH of turkey thigh muscles stored in refrigerated conditions at 1 °C for 6 days.

Ensuring the high quality and safety of meat, a product susceptible to spoilage, is essential for all participants in the food supply chain, from producers and distributors to retailers and consumers, to protect public health, avoid economic losses, and maintain consumers’ trust.

## 2. Materials and Methods

### 2.1. Materials

The animals were housed on the same agricultural establishment under uniform environmental circumstances.

The experimental material consisted of sections of thigh muscles (devoid of skin and bones, with an average mass of ±0.5 kg) obtained 24 h post-mortem from turkeys slaughtered in industrial settings. Following random selection, samples were enclosed in plastic pouches, fitted with cooling elements, and conveyed via refrigerated means from the slaughterhouse. Certain samples were promptly dispatched for analysis, whereas others were enclosed in polyamide–polyethylene foil bags. The duration from the turkeys’ slaughter to the commencement of storage and analysis was roughly 24 h (referred to as day 0). Muscles stored in plastic bags were kept in a refrigerator at +1 °C with automated temperature regulation. Testing was conducted 24 h post-slaughter (day 0) and continued for 6 consecutive days (with three muscles tested during each period). This experiment was iterated five times, encompassing a total examination of 105 muscles, with 15 for each storage duration. Determinations were conducted for each individual muscle.

### 2.2. Microbiological Analyses

Microbiological examinations encompassed the quantification of the total mesophilic aerobes, presumptive lactic acid bacteria and Enterobacteriaceae in 1 g of the specimen under investigation. The assessments were conducted in compliance with the relevant international standards [31,32,33]. The results were reported as colony-forming units per gram of sample (CFU/g) and expresses as a mean ± standard deviation.

For each meat sample, 10 g were taken, added to 90 mL of sterile peptone water, and homogenized. Serial tenfold dilutions were prepared, and 100 µL of each dilution was inoculated onto growth media. Total mesophilic aerobes were determined on plate count agar (Merck, Burlington, MA, USA) and plates were incubated at 30 °C for 72 h. Enterobacteriaceae were counted using overlay treatment on violet red bile dextrose agar (VRBD, Merck). Petri dishes were incubated for 48 h at 30 °C. De Man, Rogosa and Sharpe agar (MRS, Oxoid, Basingstoke, UK) was used to count presumptive lactic acid bacteria using the pour plate technique. Plates were incubated aerobically at 37 °C for 72 h.

### 2.3. Total Pigments Determination

The entire pigment analysis involved the extraction of total heme pigments utilizing the Warris method [34], which was then modified [35] and expressed in mg/g of meat. The muscle samples were promptly frozen at −18 °C for a duration of 24 h, followed by slicing them into thin flakes without prior thawing. Approximately 10 g of these samples were then homogenized with 50 cm^3^ of phosphate buffer (pH 6.8) at a temperature range of 4–6 °C for precisely 1 min at a speed of 3000 rpm. Subsequently, the homogenate was refrigerated at 4–6 °C for an hour. After this incubation, the homogenate underwent centrifugation at 4000× *g* for a duration of 10 min. The resulting supernatant was decanted, while the residual was subjected to an additional extraction using 42.5 cm^3^ of the aforementioned buffer, followed by centrifugation under the same conditions as before. The two supernatants were meticulously combined, and the total volume was determined. This extract was then centrifuged at 30,000× *g* for an hour and filtered using Whatman 1 paper filters. To determine the content of Mb, MbO_2_, and MMb, the procedure given by [36] was used. The absorbance levels were measured at wavelengths of 525, 545, 565, and 572 nm. To conduct this measurement, the Hewlett Packard’s diode array UV/VIS 8452 spectrophotometer was used. Subsequently, the concentrations of TP, Mb, MbO_2_, and MMb (expressed in mg per 1g of tissue), along with the relative concentrations (%) of Mb, MbO_2_, and MMb were calculated from the following relationships:Mb [%] = (0.369·A1 + 1.140·A2 − 0.941·A3 + 0.015)·100
MMb [%] = (0.777·A2 +0.800·A3 − 2.514·A1 + 1.098)·100
MbO_2_ [%] = 100 − (Mb + MMb),
where A1 = A572/A525; A2 = A565/A525; and A3 = A545/A525.

### 2.4. Color Determination

The color of the turkey muscles was assessed utilizing the spectrophotometric technique within the CIE L*a*b* color model [37]. The measurements were conducted using the Minolta CR 310 chromameter (Minolta Camera Co., Ltd., Osaka, Japan), with an illuminant D65 and a 50 mm viewing aperture. Prior to the measurements, the chromameter was calibrated using a white plate (Y = 93.50; x = 0.3114; y = 0.3190). Similarly, the colorimetric properties of the thigh muscles of turkeys were evaluated 30 min subsequent to unpacking. The outcomes were represented in terms of L* (lightness), a* (redness), and b* (yellowness), derived from the mean of five arbitrary readings on the internal section of each muscle, under ambient conditions (20 °C).

### 2.5. Colour Sensory Assessment

The sensory assessment was conducted to evaluate the color of turkey thigh muscles in a sensory laboratory, following all necessary criteria outlined in the international standards [38]. The assessment focused on the inner aspect of the muscle, involving seven well-trained assessors. Utilizing a six-point hedonic scale, where one point represents the lowest and six points signify the highest evaluation, the color was meticulously examined (refer to Table 1). The color intensity was quantified using conventional units [CU] as detailed by [39,40]. The scales were developed based on the criteria given in the [41,42] standards.

### 2.6. pH Determination

The pH value was analyzed utilizing a pH meter N-517 manufactured by Mera-Elwro, equipped with a dagger electrode type OSH-01 from Metron, which was inserted into the meat sample following the guidelines of the standard [43].

### 2.7. Statistical Analysis

The outcomes underwent statistical analysis using the Statistica computer program, version 13.1 [44]. Calculation of the means and standard deviations was performed. A one-way analysis of variance (ANOVA) was performed to establish the significance of the variations. In order to assess the significance of distinctions among the mean values, a Dunkan’s test based on the studentized range was conducted, with a significance level of *p* ≤ 0.05.

The Pearson linear correlation coefficient was utilized for the computation of the linear correlation between variables.

The analyses were carried out in triplicate, and results were expressed as a mean ± standard deviation.

## 3. Results

### 3.1. Microbiological Results

The results of the viable counts are reported in Table 2.

For the total mesophilic aerobes, statistically significant differences (*p* ≤ 0.05) emerged on the third day of storage, with counts up to 1.82 × 10^7^ CFU/g at 6 days of storage. Regarding presumptive lactic acid bacteria, statistically significant differences (*p* ≤ 0.05) emerged on the fourth day of storage, with counts up to 1.00 × 10^4^ CFU/g at 6 days of storage. As for the Enterobacteriaceae, statistically significant differences (*p* ≤ 0.05) emerged on the third day of storage, with counts up to 1.87 × 10^5^ CFU/g at 6 days of storage.

### 3.2. Heme Pigments

The storage time had a significant effect (*p* ≤ 0.05) on the total pigment (TP) content in the meat during storage (Table 3). The TP concentration decreased (*p* ≤ 0.05) gradually in the muscles up to the 6th day of storage, respectively by 69.80%. The TP concentration after 24 h amounted to 2.45 mg/g, however on the 6th day of storage, it was 0.74 mg/g (Table 3). The significant effect (*p* ≤ 0.05) on the total pigment concentration was noticed on the 3rd day.

The storage time had an influence (*p* ≤ 0.05) on the relative concentrations (%) and concentration (mg/1 g of tissue) of three myoglobin forms (MbO_2_, Mb and MMb) in the turkey meat (Table 3). The decrease (*p* ≤ 0.05) in the relative concentration of oxymyoglobin and increase in the relative concentration of myoglobin and metmyoglobin in relation to values denoted in the control sample was noticed on the 4th day (Table 3). The concentration of three myoglobin forms in the meat decreased gradually within 6 days of storage, respectively, by 61.90% for MMb, 65.78% for Mb and 74.29% for MbO_2_. The decrease (*p* ≤ 0.05) in Mb, MbO2 and MMb in the meat was noticed on the 3rd day in relation to values denoted in the control sample (Table 3).

### 3.3. Color Stability and pH

The storage time influenced (*p* ≤ 0.05) all the color parameters (L*, a*, b*) of samples during storage (Table 4). During cold storage, the L* and a* parameters decreased gradually in the muscles from initial values within 6 days of storage by 16.15% and 32.38%, respectively. In the control sample 24h after slaughter, lightness and redness amounted to 46.68 and 17.20, respectively. The significant decrease (*p* ≤ 0.05) in the L* and a* parameters was observed in the muscles on the 3rd day (Table 4). The b* values of the investigated samples increased (*p* ≤ 0.05) on the 3rd day, in comparison to control samples. On the 6th day, the values of yellowness rose by 157.24% (Table 4).

The sensory evaluation (SE) of the surface meat color intensity decreased with storage up to the 6th day (Table 4). The color of the control samples was recognized as intense pink-red (5.47 CU; Table 4). The deterioration (*p* ≤ 0.05) of the color samples was observed on the 3rd day of storage. The color of the turkey meat was still pink-red, but less intense in comparison to control samples. On the 6th day, the meat color was strongly changed to creamy—brown (2.10 CU; Table 4).

The pH of the turkey muscles increased up to the 6th day of storage (Table 4). A significant increase of pH in the muscles was noticed on the 2nd day. At the end of storage, the pH value of the meat was 6.43 (Table 4).

Based on the calculated correlation indices (Table 5), it was noticed that the pH values positively correlated with all measured bacterial types (TMA: 0.76, LAB: 0.85, EB: 0.73). At the same time, a strong negative correlation was demonstrated between meat pH and the color parameters L* and a* (L*: −0.90, a*: −0.88; Table 5).

In contrast to the color parameters L* and a*, pH positively correlated with the color parameter b* (0.88; Table 5).

A strong negative correlation (−0.86; Table 5) was found between the pH values of the meat and the sensory color assessment results.

## 4. Discussion

The present study observed an increase in the counts of total mesophilic aerobes (including Enterobacteriaceae) over time. Among the most reported microorganisms occurring in raw meat, those belonging to the Enterobacteriaceae family (e.g., *Escherichia*, *Klebsiella*, *Salmonella*, *Shigella*, and *Yersinia*), *Campylobacter*, *Listeria monocytogenes*, and spoilage microorganisms such as *Acinetobacter*, *Brochothrix*, *Pseudomonas*, and *Psychrobacter* were found [45]. Of note, despite the low storage temperature maintained during the 6-day monitoring, most of the occurring microorganisms were able to proliferate. Indeed, although the optimal growth temperature of most of the species harboring on the raw meat is between 30 and 37 °C, an ample range of variation in the growth speed must be considered. Hence, it is likely that, although at a reduced speed, many of the taxa were able to grow during refrigeration [46].

Regarding lactic acid bacteria, although considered pro-technological microorganisms, members of this microbial group can be the causative agents of raw meat spoilage due to acidification or production of exopolysaccharides (EPS), with the subsequent slimy texture of the meat surface [47]. Lactic acid bacteria are generally considered mesophilic microorganisms, although many species can also be moderately psychotropic [48], thus explaining the increase in the counts over time that occurred in the samples under study. Interestingly, the use of lactic acid bacteria as natural agents of meat biopreservation has been proposed [49]. Indeed, species of this microbial group can produce antimicrobial compounds known as bacteriocins. However, the most suitable species of lactic acid bacteria has to be chosen (e.g., species with low production of organic acids or EPS) to avoid unwanted modifications of the meat matrix.

As a general consideration, it is noteworthy that the initial load of microorganisms in the raw meat strongly influences the effectiveness of the refrigerated storage at the end of the meat’s shelf-life. Hence, applying good manufacturing practices could help reduce microbial loads.

The microbiological limit for aerobic mesophilic microorganisms in chilled meat, as established by the European Commission under regulation (EC) No 2073/2005 [50] and American standards is 5 × 10^5^ CFU/g [50,51]. The total mesophilic aerobic counts in meat samples should not exceed this limit to ensure the microbiological safety and quality of the product [52]. Therefore, it can be assumed that the fourth day of storage was the last for the supposed shelf life of meat stored at 1 °C (Table 2).

The accumulation of metabolites from the growth of various microorganisms can indeed impair the color of meat. Metabolomics studies have shown that color-stable muscles have higher levels of glycolytic metabolites, which can be affected by spoilage microorganisms [53]. Their metabolic activity can also produce compounds responsible for discoloration, off-odors, and off-flavors [54,55]. Specific metabolites, such as NADH and glutamate, improved color stability, while methionine had detrimental effects [56]. The presence of aerobic mesophilic microorganisms, including Enterobacteriaceae, has been associated with the spoilage of food products, indicating their potential to affect the color of poultry meat [57]. In the present study, the activity of microorganisms likely resulted in a decrease in the color parameters L* and a* and an increase in the color parameter b* during the storage of turkey muscles (Table 4). Several research studies have demonstrated that specific bacteria have the ability to produce pigments through their metabolic activities, resulting in changes to the natural color of meat, such as an enhancement in its yellowness [58,59]. Furthermore, modification in the color of meat, which includes a rise in yellowness, has been linked to bacterial spoilage, myoglobin autoxidation, and protein oxidation [60].

During the storage of poultry meat, the growth of putrefactive microflora has the potential to cause an increase in pH levels in the surrounding environment as a result of protein breakdown and amino acid decarboxylation [61,62,63,64]. The rise in volatile basic nitrogen levels in meat as it is stored is linked to the process of amino acid deamination, which generates ammonia [65]. Bacteria can decompose meat amino acids into methyl sulfides, esters, and acids, accumulating these compounds and potentially causing an increase in pH [66,67].

In the present study, the increase of pH in the muscles from the 2nd day could be associated with an increase in the number of determined microorganisms in stored meat. Research indicates that a variety of bacteria, including Enterobacteriaceae, can produce alkaline compounds such as ammonia [68,69,70], which leads to the progressive alkalinization of stored muscle, and thus to the increase of pH. The pH of meat, undergoing an increment throughout its storage duration, can significantly impact the functioning of enzymes responsible for the degradation of hemoproteins like myoglobin. The process of alkalinization can influence the steadiness and alteration of various myoglobin configurations, such as deoxymyoglobin, oxymyoglobin, and metmyoglobin, which play a critical role in determining the color and visual appeal of the meat. The heightened pH level promotes the transformation of oxymyoglobin into metmyoglobin, thereby contributing to a less desirable brown color in the meat [65]. Hoa et al., 2020 [71] observed a decline in OxyMb levels with escalating acidity levels, indicating a probable surge in enzyme performance. This assertion was corroborated by Tushar et al., 2023 [12], who shed light on the pivotal role of pH in the reduction of metmyoglobin, a process directly influencing the stability of meat color. Ragucci et al., 2021 [72] drew attention to this phenomenon by noting an enhancement in pseudoperoxidase activity in the presence of Ca^2+^ at a pH of 5.8, hinting at a plausible association between pH levels and enzyme efficacy.

It was discovered that as the pH increased, the brightness of the meat decreased, and it became darker (Table 5). The research results are consistent with the findings of other researchers, who indicate that meat with a higher pH tends to be darker in color, as indicated by a lower L* value [73,74,75,76,77].

### Study Limitations

This study was conducted under controlled laboratory conditions, which may not accurately reflect the variability in real-world commercial storage environments. This includes consistent refrigeration at 1 °C, which might not be maintained in practical settings, affecting the generalizability of the results. The investigation primarily focused on microbial growth, color stability, and pH. Other critical quality parameters, such as, e.g., texture and flavor, were not assessed. Including these parameters in future studies would provide a more holistic view of the factors influencing turkey meat quality during storage.

## 5. Conclusions

Based on the conducted research, it was found that the maximum storage time of turkey thigh muscles at a temperature of 1 °C is 4 days. Up to the fourth day of storage, the turkey thigh muscles maintained an acceptable quality regarding microbial growth, color stability, and pH levels. By this time, the total heme pigment content decreased from 2.45 mg/g on day 0 to 1.59 mg/g, but the reduction was not significant enough to adversely affect the meat’s color quality. The lightness decreased slightly from 46.68 to 44.02, and the redness decreased from 17.20 to 16.18, within acceptable consumer preference ranges. The yellowness increased from 3.04 to 5.03, indicating some changes but not enough to negatively impact the overall sensory evaluation, which remained relatively high. The authors noticed a strong correlation between the values of color parameters L*, a*, and b* and the meat’s sensory assessment of color (SE). Therefore, color parameters may indicate the consumer desirability of turkey muscles during storage. The microbial counts remained within acceptable limits up to the fourth day. Total mesophilic aerobes increased from 1.20 × 10^3^ CFU/g to 3.5 × 10^5^ CFU/g. These levels are below the thresholds typically associated with spoilage, indicating that the meat remained safe for consumption. The pH of the turkey muscles increased from 5.94 to 6.12 by the fourth day, a change that was significant but not sufficient to cause spoilage.

It should be noted that although the tests were carried out at a controlled temperature of 1 °C, it is not certain whether similar conditions are maintained during the commercial storage of meat, which may affect the actual safety and quality of the product available to consumers. The results of this research are aimed to raise consumer awareness and optimize storage conditions of producers, which may contribute to reducing food waste and improving production efficiency. This study, therefore, provides valuable data that can support informed purchasing decisions and improve storage practices for meat products in consumers’ homes.

## Figures and Tables

**Table 1 microorganisms-12-01114-t001:** Criteria of a six-point scale of the sensory assessment of color of turkey thigh meat.

Points	Color
6	Ideal, typical (intense pink-red)
5	Typical (pink-red, even)
4	Typical but less intense (pink and red)
3	Slightly changed (noticeably changed in places—lighter or darker pink and red)
2	Strongly changed (light, creamy or very dark—brown)
1	Completely changed, putrid (gray, brown, blue)

**Table 2 microorganisms-12-01114-t002:** Development of bacteria in the chilled stored turkey thigh muscles.

	Storage Time						
Bacteria Type (CFU/g)	0* Day	1 Day	2 Days	3 Days	4 Days	5 Days	6 Days
Total mesophilic aerobes	1.20 × 10^3 a^	1.41 × 10^3 a^	2.34 × 10^3 a^	2.35 × 10^4 b^	3.5 × 10^5 c^	1.50 × 10^6 d^	1.82 × 10^7 e^
Presumptive lactic acid bacteria	1.01 × 10^2 a^	1.47 × 10^2 a^	2.57 × 10^2 a^	8.94 × 10^2 a^	1.10 × 10^3 b^	1.60 × 10^3 b^	1.00 × 10^4 c^
Enterobacteriaceae	4.36 × 10 ^a^	7.00 × 10 ^a^	8.08 × 10 ^a^	2.65 × 10^2 b^	1.08 × 10^3 c^	2.08 × 10^4 d^	1.87 × 10^5 e^

0* day (24 h after slaughter). The data are averaged values of 15 tests; a, b, c, d, e—values with different letters in the same row differ at *p* ≤ 0.05.

**Table 3 microorganisms-12-01114-t003:** Heme pigment concentration in the chilled stored turkey thigh muscles.

Storage Time	TP	Mb		MbO_2_		MMb	
	C	RC	C	RC	C	RC	C
0*	2.45 ^a^ ± 0.27	0.31 ^a^ ± 0.01	0.76 ^a^ ± 0.09	0.42 ^a^ ± 0.05	1.05 ± 0.17	0.26 ^a^ ± 0.05	0.63 ^a^ ± 0.11
1 day	2.44 ^a^ ± 0.27	0.32 ^a^ ± 0.01	0.78 ^a^ ± 0.11	0.44 ^a^ ± 0.02	1.06 ± 0.13	0.25 ^a^ ± 0.03	0.60 ^a^ ± 0.08
2 days	2.36 ^a^ ± 0.20	0.32 ^a^ ± 0.02	0.76 ^a^ ± 0.08	0.41 ^a^ ± 0.03	0.97 ± 0.11	0.26 ^a^ ± 0.02	0.64 ^a^ ± 0.07
3 days	1.85 ^b^ ± 0.11	0.32 ^a^ ± 0.01	0.60 ^b^ ± 0.03	0.42 ^a^ ± 0.02	0.78 ± 0.07	0.26 ^a^ ± 0.02	0.48 ^b^ ± 0.04
4 days	1.59 ^c^ ± 0.09	0.34 ^b^ ± 0.01	0.54 ^b^ ± 0.03	0.37 ^b^ ± 0.03	0.59 ± 0.06	0.29 ^b^ ± 0.03	0.46 ^b^ ± 0.04
5 days	1.31 ^d^ ± 0.11	0.35 ^b^ ± 0.03	0.46 ^c^ ± 0.06	0.35 ^b^ ± 0.04	0.46 ± 0.06	0.29 ^b^ ± 0.03	0.38 ^c^ ± 0.06
6 days	0.74 ^e^ ± 0.13	0.34 ^b^ ± 0.01	0.26 ^d^ ± 0.03	0.35 ^b^ ± 0.02	0.27 ± 0.04	0.31 ^b^ ± 0.02	0.24 ^d^ ± 0.03

0* day (24 h after slaughter). The data are the averaged values of 15 tests (±standard deviation). TP—Total pigment concentration; Mb—myoglobin; MbO_2_—oksymyoglobin; MMb—metmyoglobin; C—concentration mg/1 g of tissue; RC—relative concentration; a, b, c, d, e—values with different letters in the same column differ at *p* ≤ 0.050.

**Table 4 microorganisms-12-01114-t004:** Evaluation of the turkey thigh muscles’ color and pH values.

Variables	Storage Time						
	0* Day	1 Day	2 Days	3 Days	4 Days	5 Days	6 Days
L*	46.68 ^a^ ± 1.30	46.54 ^a^ ± 1.52	46.34 ^a^ ± 0.54	45.19 ^b^ ± 1.12	44.02 ^c^ ± 1.23	43.24 ^c^ ± 0.98	39.14 ^d^ ± 1.52
a*	17.20 ^a^ ± 0.95	17.16 ^a^ ± 0.36	17.04 ^a^ ± 0.38	16.23 ^b^ ± 0.68	16.18 ^b^ ± 0.46	14.22 ^c^ ± 0.52	11.63 ^d^ ± 0.37
b*	3.04 ^a^ ± 0.22	3.08 ^a^ ± 0.52	3.38 ^a^ ± 0.52	3.44 ^b^ ± 0.39	5.03 ^c^ ± 0.28	6.11 ^d^ ± 0.33	7.82 ^e^ ± 0.53
SE [CU]	5.47 ^a^ ± 0.19	5.30 ^a^ ± 0.80	5.22 ^a^ ± 0.22	4.70 ^b^ ± 0.70	3.62 ^c^ ± 0.27	3.02 ^d^ ± 0.21	2.10 ^e^ ± 0.12
pH	5.94 ^a^ ± 0.07	5.95 ^a^ ± 0.07	6.01 ^b^ ± 0.05	6.08 ^c^ ± 0.06	6.12 ^c^ ± 0.09	6.25 ^d^ ± 0.10	6.43 ^e^ ± 0.05

0* day (24 h after slaughter). The data are the averaged values of 15 tests (±standard deviation); SE—sensory evaluation of color; a, b, c, d, e—values with different letters in the same row differ at *p* ≤ 0.05.

**Table 5 microorganisms-12-01114-t005:** Correlation coefficients between variables.

	TMA	LAB	EB	TP	MbC	MMbC	MbO_2_C	pH	L*	a*	b*	SE
TMA	1.00	0.92	0.93	−0.70	−0.74	−0.58	−0.68	0.76	−0.85	−0.88	0.80	−0.73
LAB	0.92	1.00	0.95	−0.79	−0.82	−0.67	−0.77	0.85	−0.85	−0.92	0.87	−0.80
EB	0.93	0.95	1.00	−0.66	−0.68	−0.54	−0.65	0.73	−0.81	−0.85	0.79	−0.69
TP	−0.70	−0.79	−0.66	1.00	0.99	0.89	0.96	−0.89	0.86	0.89	−0.91	0.93
MbC	−0.74	−0.82	−0.68	0.99	1.00	0.88	0.95	−0.89	0.87	0.91	−0.92	0.92
MMbC	−0.58	−0.67	−0.54	0.89	0.88	1.00	0.75	−0.78	0.76	0.79	−0.78	0.82
MbO_2_C	−0.68	−0.77	−0.65	0.96	0.95	0.75	1.00	−0.86	0.82	0.84	−0.90	0.91
pH	0.76	0.85	0.73	−0.89	−0.89	−0.78	−0.86	1.00	−0.90	−0.88	0.88	−0.86
L*	−0.85	−0.85	−0.81	0.86	0.87	0.76	0.82	−0.90	1.00	0.96	−0.90	0.92
a*	−0.88	−0.92	−0.85	0.89	0.91	0.79	0.84	−0.88	0.96	1.00	−0.93	0.91
b*	0.80	0.87	0.79	−0.91	−0.92	−0.78	−0.90	0.88	−0.90	−0.93	1.00	−0.94
SE	−0.73	−0.80	−0.69	0.93	0.92	0.82	0.91	−0.86	0.92	0.91	−0.94	1.00

TMA—total mesophilic aerobes (CFU/g); LAB—presumptive lactic acid bacteria; EB—Enterobacteriaceae; TP—total heme pigment concentration (mg/g); MbC—myoglobin concentration; MetC—metmyoglobin concentration; MbO_2_C—oxymyoglobin concentration; L*—lightness; a*—redness; b*—yellowness; SE—sensory evaluation of color; correlation coefficients marked in red differ at *p* ≤ 0.05.

## Data Availability

The original contributions presented in the study are included in the article, further inquiries can be directed to the corresponding author.

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
