# Peer review of "Impact of Refrigerated Storage on Microbial Growth, Color Stability, and pH of Turkey Thigh Muscles"

_microorganisms, 2024, doi:10.3390/microorganisms12061114_

Round 1

Reviewer 1 Report

Comments and Suggestions for Authors

The authors investigated a topic that is always current, and it refers to changes in meat during storage in the refrigerator.

The introduction provides enough information about the issue that the paper deals with.

The chosen research methods are correct. My query refers to selected microbiological methods that are related to Polish standards (references 25-27). Are these methods compliant with ISO methods?

The results are presented in tables that are readable and can be interpreted, and then discussed with the results of similar research.

The chosen literature is correct.

Author Response

Thank you for doing the review. Answer in the attachment.

Reviewer 2 Report

Comments and Suggestions for Authors

The study investigated the effect of refrigerated storage on the microbiological contamination, color, and pH of turkey thigh muscles stored at 1℃ over six days. The results revealed that the maximum storage time for turkey thigh muscles at 1℃ was four days. This study underscores the critical necessity of maintaining controlled refrigeration conditions to reduce spoilage, ensure food safety, and preserve the sensory and nutritional qualities of turkey meat. However, several issues need to be addressed in this paper:

1. The introduction section is not focused and the authors should have stated the framework of the study in this section.

2. The selection process for the 1g samples in section 2.2.2 should be detailed, as microbial content varies across different regions of a turkey thigh.

3. Presenting the data in table format may not effectively reflect changes in indicators; it is suggested to use graphs instead.

4.In line 219, the authors fail to provide explanations for the changes of different microbial.

5.In line 235, the reasons for changes in myoglobin and other pigments are not clarified.

6.Why no blank control experiment?

7.In line 273, the authors state that with an increase in bacterial population, the pH of stored turkey meat also increases, but there is no supporting data provided. Authors are encouraged to supplement with relevant experimental data.

8. In line 35, the authors mention that the initial microbial load in raw meat significantly affects the efficacy of refrigeration during shelf life, but relevant experiments were not conducted. Supplementary experiments would enhance the paper's completeness.

9. The discussion section should be streamlined, as some content repeats earlier sections.

10. The title and conclusion need adjustment for better.

11. Some citations are not formatted correctly.

Comments on the Quality of English Language

Minor editing of English language required

Author Response

(The authors gave the same response as above.)

Reviewer 3 Report

Comments and Suggestions for Authors

Please, provide the most relevant results (values) in the Abstract. In this section, it is also recommended to include some directions for future investigations.

Lines 36-38: References are missing.

Lines 39-41: References are missing.

Lines 41-43: References are missing.

Lines 69-74: References are missing.

What is the novelty of this study? A lot of work is described in the literature on this topic. The authors need to clearly justify the need to carry out this investigation in the Introduction section and present its novelty compared to other similar papers.

Line 126: Why Polish Standards and not international ones?

Why some values are highlighted in red in Table 5? Please, provide an explanation for this.

Some sentences in the Results section are truly discussions of the same. I advise the authors to standardize this point, perhaps by unifying the two sections or limiting the Results section to the results themselves. See the examples of Lines 285-289; Lines 295-299.

Lines 308-312: This is not a discussion of the results and should be removed from this section.

Lines 340-341: Why did you compare with ANVISA and not with European Food Legislation, mentioning the European Food Safety Authority?

Study limitations should be pointed out at the end of the Discussion section.

What is the relevance of this study and its conclusions, considering current knowledge and the vast literature that already exists on this topic? This is a fundamental issue that must be reflected in the manuscript, to justify its need for publication in Microorganisms.

Comments on the Quality of English Language

Minor editing of English language required.

Author Response

(The authors gave the same response as above.)

Round 2

Reviewer 2 Report

Comments and Suggestions for Authors

accept

Comments on the Quality of English Language

Minor editing of English language required

Reviewer 3 Report

Comments and Suggestions for Authors

The authors revised the manuscript properly. In my point of view, it can be accepted for publication in Microorganisms.